# GENERALIZING REINFORCEMENT LEARNING TO UNSEEN ACTIONS

## ABSTRACT

A fundamental trait of intelligence is the ability to achieve goals in the face of novel circumstances. In this work, we address one such setting which requires solving a task with a novel set of actions. Empowering machines with this ability requires generalization in the way an agent perceives its available actions along with the way it uses these actions to solve tasks. Hence, we propose a framework to enable generalization over both these aspects: understanding an action's functionality, and using actions to solve tasks through reinforcement learning. Specifically, an agent interprets an action's behavior using unsupervised representation learning over a collection of data samples reflecting the diverse properties of that action. We employ a reinforcement learning architecture which works over these action representations, and propose regularization metrics essential for enabling generalization in a policy. We illustrate the generalizability of the representation learning method and policy, to enable zero-shot generalization to previously unseen actions on challenging sequential decision-making environments. More training and testing videos can be found at `sites.google.com/view/action-generalization/`

## 1 INTRODUCTION

Imagine visiting your friend for the first time, and you decide to cook your favorite dish there. But since you have never been in their kitchen before, there could be certain tools you have never seen, like an odd-shaped sponge. However, by looking at its porous texture or observing its interaction with water, you can understand that this object can absorb liquid. Later during cooking when you want to clean the table, you can select that sponge since you can relate its absorbing characteristics with another tool you have used for cleaning. Just like in this scenario, our tasks often involve making selections from novel or unseen entities. When we encounter such choices, we examine them to first understand their functionality which informs our selection process while solving a task.

Can machines also understand previously unseen choices and subsequently use them for solving tasks? From a reinforcement learning perspective, this brings an interesting question of how to enable generalization of discrete action policies to solve tasks using unseen sets of actions. Prior work in deep reinforcement learning has explored generalization over environments (Cobbe et al., 2018; Nichol et al., 2018), and tasks (Finn et al., 2017; Parisi et al., 2018). However, action space generalization is relatively unexplored and is crucial for agents to be flexible in the face of novel circumstances, like selecting an unseen sponge for a known task of cleaning in above example.

In this work, our goal is to develop a framework that reflects the two phases of solving action generalization: (1) general understanding of unseen discrete actions from their characteristic information (like appearance or behaviors), and (2) training a policy to solve tasks by utilizing this general understanding. However, an action can have diverse behaviors and hence requires a collection of data (e.g. different viewpoints, videos or state trajectories of how it effects on environment) to sufficiently express this diversity. Hence, the primary challenge is to develop a generalizable unsupervised learning method which can extract an action's characteristics from a dataset constituting its diverse effects. To this end, we propose to embed actions' datasets by extending the work on hierarchical variational autoencoders (Edwards & Storkey, 2017).

The obtained embeddings reflect an action's general utility, and can be used as action representations in the downstream task of reinforcement learning. However, conventional reinforcement learning

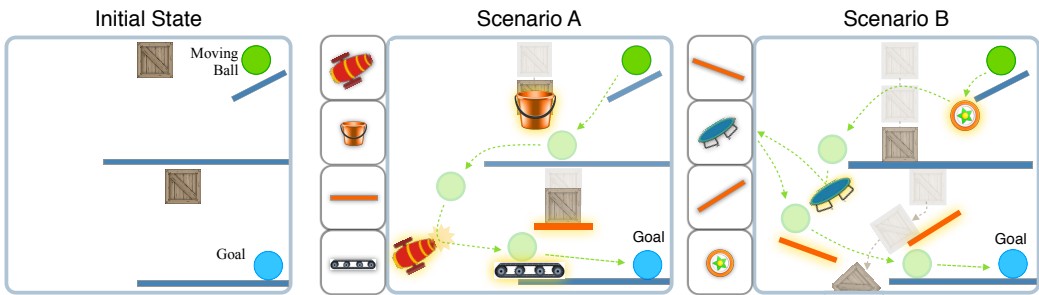

(a) Chain Reaction Tool Environment (CREATE)

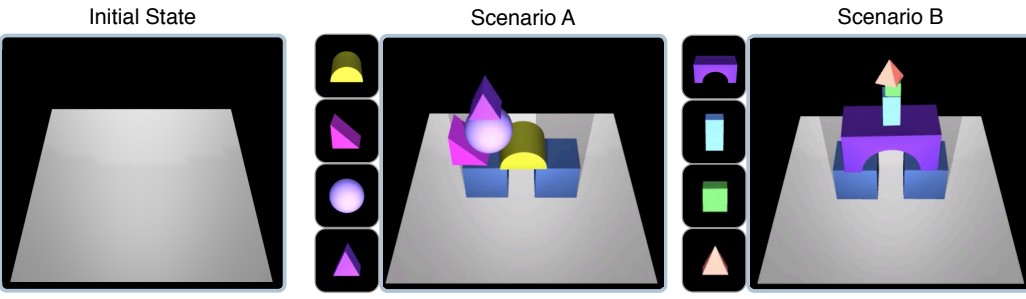

(b) Shape Stacking

Figure 1: Generalizing the knowledge of solving a task to a new set of actions. (a) CREATE is a sequential environment where the task is to help the green ball reach the goal (blue) by selecting tools and deciding where to place them. (b) In Shape Stacking the goal is to stack a tall tower by selecting the right shapes and their placements. Scenario A depicts the training scenario when the agent learns to utilize a given set of actions to solve the task. Scenario B presents an unseen set of actions to the agent which is expected to generalize to solve the task zero-shot.

algorithms utilize the available actions in a way that best optimizes a reward. This directly incentivizes a policy to overfit to the actions seen during training, just like the problem of overfitting to training data in supervised learning. To address this challenge, we formulate this problem as risk minimization (Vapnik, 1992) for reinforcement learning, and propose regularization objectives to enforce generalization of policy to unseen actions.

The main contributions of this paper are: (1) introducing the problem and a proposed solution to enable action space generalization in reinforcement learning, (2) representing an action with a dataset reflecting its diverse characteristics, and employing a generalizable unsupervised learning approach to embed these datasets. (3) a method to use learned action representations in reinforcement learning, and regularization methods to enable learning of generalizable policies.

## 2 RELATED WORK

**Generalization in reinforcement learning** In typical deep reinforcement learning (RL) settings (Mnih et al., 2015; 2016; Lillicrap et al., 2015; Schulman et al., 2017), a policy or value network learns to act over an action space of fixed dimensionality. By taking states or observations as input to neural networks, these methods are able to generalize to unseen environment states drawn from a similar distribution as training (Cobbe et al., 2018; Nichol et al., 2018). Similarly, prior works have explored generalization in RL for unseen instructions (Oh et al., 2017), new sequences of subtasks (Andreas et al., 2017), manipulation of unseen tools (Fang et al., 2018; Xie et al., 2019), task demonstrations (Xu et al., 2017), and agent morphologies (Wang et al., 2018; Sanchez-Gonzalez et al., 2018; Pathak et al., 2019). In contrast, our framework enables zero-shot generalization of RL policies when the agent gets a previously unseen action set.

**Unsupervised representation learning for downstream tasks** Bengio et al. (2013) state representation learning of data makes it easier to extract useful information when building predictors. Prior

works show that such representations have been useful for a variety of downstream tasks, like classification and video prediction (Denton et al., 2017), visually representing objects for relational reasoning tasks (Steenbrugge et al., 2018), representing image-states for domain adaptation in RL (Higgins et al., 2017), and, representing goals for better exploration (Laversanne-Finot et al., 2018) and sample efficiency (Nair et al., 2018) in RL. In this paper, we show how unsupervised representation learning over datasets (Edwards & Storkey, 2017) can be used for embedding discrete actions, and enable generalization in the downstream task of reinforcement learning.

**Action Representations** Using continuous representations of discrete actions, a policy can be trained through a combined Q-function over state and action representations (He et al., 2015), or in an actor-critic architecture by selecting the nearest neighbor action vector to the policy's continuous output (Van Hasselt & Wiering, 2009; Dulac-Arnold et al., 2015). Unlike our work, these prior works assume access to ground truth action representations, which are usually not readily available. In other related work, action representations are learned implicitly through inverse model on a fixed action space to ease learning in large discrete action spaces (Chandak et al., 2019) or for intrinsic reward (Kim et al., 2019). In contrast, we do not have the assumption of fixed action space as we learn action representations separately, and hence are able to incorporate new actions for the same policy. While Tennenholtz & Mannor (2019) pre-learn action representations explicitly using co-occurrence of actions in task-specific demonstrations, our generic embedding method applies to various modalities of datasets to represent actions, which are task-independent and hence suited for generalization to unseen actions.

**Skill and Trajectory Embeddings** In reinforcement learning, variational autoencoders (VAE) (Kingma & Welling, 2014) are often used for learning an abstraction for continuous entities like skills and state-action trajectories. Specifically, Co-Reyes et al. (2018) utilize a trajectory autoencoder for hierarchical RL, and Lynch et al. (2019) learn a latent space of trajectories and employ a goal-conditioned planner over it. Hausman et al. (2018) learn an embedding space of skills through a shared policy for different tasks, and utilize this space for solving other related tasks. In this paper, we extend the framework of hierarchical VAE (Edwards & Storkey, 2017; Achille et al., 2019) to trajectories, so as to embed even sequential datasets which are better indicative of action behavior. In general, an action can be a discrete skill choice, and an action's behavior can be represented as the trajectory of effects it causes on the environment. Since individual trajectories are incapable of capturing the diverse effects of actions, we propose to use datasets for representing actions.

## 3 GENERALIZATION TO UNSEEN ACTIONS

Our approach is based on the intuition that when humans encounter previously unseen discrete entities, we examine them to understand their functionality through visual inspection or physical interaction, before deciding what to select for a task. Once the general functionality is inferred, these discrete objects can be used as actions in decision-making tasks, like selecting a tool for cooking or furniture assembly. In this paper, we incorporate these two phases (Figure 2) to enable agents to utilize previously unseen actions: (1) extracting representations of actions from datasets of unstructured information (e.g. image, videos), and (2) training a reinforcement learning policy to utilize these action representations with the joint objective of generalization and reward maximization.

In order to represent actions, we note that an action can have diverse behaviors like how it interacts with its environment. Further, there can be various ways an agent observes this dataset. In the sponge example, the action exhibits diverse properties like absorption or compression, and the agent can observe this through porous texture (image) or through interacting with it (states trajectory). Therefore, in its most general form, information about an action can be expressed in the form of a diverse collection of unstructured data like images, videos or trajectories. To learn action representations in an unsupervised and generalizable, we use a hierarchical VAE and extend it to sequence data like videos (Section 3.2). Next, we show how a policy is trained to use these action representations as input, and propose training objectives for enabling generalization (Section 3.3).

### 3.1 PRELIMINARIES

For a learning agent, we denote the entire set of possible discrete actions as $\mathbb{A}$. For evaluation, we assume an episodic setting, where the agent only has a subset $\mathcal{A} \subset \mathbb{A}$ of actions available to it. Each action $a \in \mathbb{A}$ has an associated dataset $D = \{\boldsymbol{x}_1, \ldots, \boldsymbol{x}_L\}$ of observable samples $x_n \sim P(\boldsymbol{x}|\boldsymbol{a})$

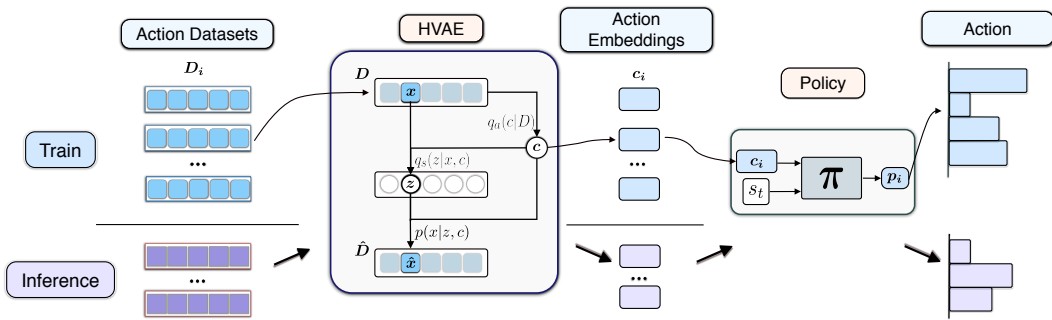

Figure 2: Framework for generalization to unseen actions. (1) Action datasets for all training actions are used to train a Hierarchical VAE (HVAE) model. (2) The action encoder embeds each dataset to define the approximate posterior $q_a(c|D)$ over action latents $c$. (3) The instance encoder $q_s(z|x,c)$ encodes each data sample $x$, while conditioned on the action latent $c$, into a distribution over instance latents $z$. (4) The decoder $p(x|z,c)$ reconstructs the action sample $x$ based on the action embedding $c$ and sample latent $z$. (5) The policy $\pi$ takes current state $s_t$ and the inferred action embeddings $c_i$ for each of the given actions and produces a categorical distribution to represent the policy. Similar flow occurs at inference, when new actions and their datasets are given.

which are characteristic of the action $a$. During training, the agent only has access to a subset $\mathcal{A}_K \subset \mathbb{A}$ of known actions. During evaluation, action set $\mathcal{A}$ can even be totally unseen for the agent.

The action set $\mathbb{A}$ constitutes the discrete action space of an episodic Markov decision process (MDP). Formally, $\{\mathcal{S}, \mathbb{A}, \mathcal{T}, R, \gamma\}$ defines the set of states, actions, transition probability, reward function, and discount factor of an MDP. Given a set of available actions $\mathcal{A} \subset \mathbb{A}$ at any time step $t$, the core problem is to learn parameters $\theta$ of a policy $\pi_\theta(a_t|s_t)$, which defines a probability distribution over actions $a_t \in \mathcal{A}$ for a state $s_t$. Since the available action sets $\mathcal{A}$ are stochastically sampled and the environments are in general stochastic, we primarily consider stochastic policies in this paper. The performance of $\pi_\theta$ is evaluated based on a discounted return $R = \sum_{t=0}^{T-1} \gamma^t r(s_t, a_t)$ where $r$ is the reward function and $T$ is the episode horizon. The aim is to train a policy which only has access to the known actions $\mathcal{A}_K$ and its datasets, but generalizes to maximize reward on unseen actions.

### 3.2 Unsupervised learning of action representations

We represent the diverse characteristics of an action with a dataset of observed information. To extract usable information from these action datasets, we propose an unsupervised representation learning method to learn action embeddings. Our key insight is that the common information underlying different samples of an action's dataset best represents the general properties of that action.

Therefore, we aim to learn an action encoder to map each discrete action's entire dataset to a continuous representation. For unsupervised learning of this encoder, we can use a variational autoencoder (VAE) with reconstruction objective (Kingma & Welling, 2014). However, since the input to VAE is in the form of a dataset, it should capture the information shared across multiple data samples. Therefore we encode both, the action datasets and the sample within each action's dataset into a hierarchy of connected latent spaces.

Such a hierarchical VAE (HVAE) architecture has been explored by Edwards & Storkey (2017) for few-shot classification and clustering of datasets. We use it for the purpose of encoding action datasets and using them for generalization (Figure 2). HVAE is composed of an action VAE over datasets and an instance VAE over samples. The encoders and decoders of the instance VAE are conditioned on its parent action latent vector. For each action $a$ and its associated dataset $D = \{x_1, \ldots, x_L\}$, the action encoder $q_a(c|D)$ is used to sample an action latent $c$, while regularized by an action prior $p_a(c)$. For each action sample $x \in D$, the instance encoder $q_s(z|x,c)$ is used to sample a latent $z$ encoding the sample instance $x$, while conditioned on $c$. The prior distribution $p_s(z|c)$ as well as the decoder $p(x|z,c)$ are also conditioned on the action latent. For each action dataset, ELBO comprises of reconstruction over data samples and the two KL divergence terms (Edwards & Storkey, 2017):

$$\mathcal{L}_D = \mathbb{E}_{c\sim q_a(.|D)}\left[\sum_{x\in D}\mathbb{E}_{z\sim q_s(.|x,c)}[\log p(x|z,c)] - \mathcal{D}[q_s(z|x,c)||p_s(z|c)]\right] - \mathcal{D}[q_a(c|D)||p_a(c)]$$
(1)

We further extend this framework to incorporate sequential data like state trajectories and videos, as that is more suitable to express behaviors of actions. For a dataset of trajectories $\tau$, we use a Bi-LSTM encoder for $q_a(c|D)$, and LSTM decoder $p(\tau|z,c,s)$ which also takes the initial state $s$ of $\tau$ and reconstructs the rest of it (Schuster & Paliwal, 1997; Wang et al., 2017; Co-Reyes et al., 2018). For the case of video datasets, we also incorporated temporal skip connections (Ebert et al., 2017) from $s$ by predicting an extra mask channel, to weigh contributions from the predicted frame and the first frame $s$.

For getting representations of any action $a \in \mathbb{A}$ (seen or unseen) through a trained HVAE, we use the action dataset encoder $q_a(c|D_a)$ output's mean as the representation $c_a$ (Figure 2). This choice of using mean as representation follows prior work like Higgins et al.; Steenbrugge et al. (2018), but one could also use sampling from the output distribution as representation, as done in Locatello et al. (2019). The generalizability of these representations to unseen actions depends on whether the action's behaviors lie in the distribution of behaviors of known actions. Hence, the hierarchy in HVAE makes it an expressive encoder for actions, since even seemingly new discrete actions can have characteristics which belong to the distribution of previously seen effects.

### 3.3 LEARNING POLICIES OVER ACTION REPRESENTATIONS

While solving tasks with new actions, humans first form a general interpretation of the behaviors of actions, and then utilize it to take appropriate actions. Similarly, once our agent learns actions representations based on observed datasets (section 3.2), it should learn to utilize them for solving tasks. This involves not only extracting the task-specific information from the representations, but also doing so in a generalizable manner so that it can utilize previously unseen action representations.

Here we assume access to an embedder $\phi$, and hence the associated action representations $c_a = \phi(D_a)$ for each $a \in \mathbb{A}$. Our aim is to learn a policy $\pi_\theta(a|s,\mathcal{A},\phi)$ which maximizes the expected reward under any set of available actions $\mathcal{A} \subset \mathbb{A}$. We propose to utilize the action representations $c_a$ as inputs to the policy, which acts as a function approximator over action representations and states. Specifically, our policy consists of a utility function $f_\theta : \mathcal{S} \times \mathbb{R}^d \to \mathbb{R}$, which maps a $d$-dimensional action embedding and a state to its utility. The probability distribution over actions is simply defined as the Softmax over the utilities of each available action $a' \in \mathcal{A}$.

$$\pi_\theta(a|s,\mathcal{A}) = \frac{e^{f_\theta(s,c_a)}}{\sum_{a'\in\mathcal{A}}e^{f_\theta(s,c_{a'})}}$$
(2)

We can train the parameters $\theta$ using policy gradient methods on $\pi$.

### 3.4 ENABLING GENERALIZATION TO UNSEEN ACTIONS

The primary objective is to find parameters $\theta$ of a policy which maximizes rewards on unseen action sets $\mathcal{A} \subset \mathbb{A}$. We formulate this generalization problem with statistical learning theory (Vapnik, 1998; 2013), and propose regularization objectives which aim to satisfy its assumptions. The theory mainly deals with generalization in supervised learning problems with an assumption on training examples to be independent and identically distributed (i.i.d. sampled). In a reinforcement learning setup with action representations $c_a$, the objective becomes minimizing the theoretical risk of the policy:

$$\min_\theta \text{Risk}(\pi_\theta) = \min_\theta \mathbb{E}_{s,c_a}[L(f_\theta(s,c_a),y^*)] = \max_\theta \mathbb{E}_{\mathcal{A}\sim\mathbb{A},a\sim\pi_\theta(.|s,\mathcal{A})}[R_{\pi_\theta}(s,c_a)]$$
(3)

Here $L$ is a real-valued loss function which measures the optimality of policy hypothesis $\pi_\theta$ (equivalently the utility function $f_\theta$) with respect to the output $y^*$ of an optimal stochastic policy $\pi^*$ at state $s$. While $L$, $\pi^*$ or $y^*$ cannot be defined in closed form, the definition of optimal policy (Sutton & Barto, 2018; Sutton et al., 2000) makes this objective equivalent to maximizing the cumulative reward $R$, given an unseen action space $\mathcal{A}$ and their action representations $c_a$. Note that the expectation in Eq. 3 is also over states $s$ drawn from environment, but dropped for readability.

During training, the agent only has access to a limited set of known actions $\mathcal{A}_K \subset \mathbb{A}$. The standard reward maximization objective in RL with training set of actions, $\mathcal{A}_K$ is equivalent to Empirical Risk Minimization (ERM) of the hypothesis $\pi_\theta$ (Vapnik, 1992). Hence, the ERM training objective is:

$$\max_\theta \mathbb{E}_{a \sim \pi_\theta(.|s, \mathcal{A}_K)}[R_{\pi_\theta}(s, c_a)] \tag{4}$$

However, a policy trained with ERM is prone to overfitting to data seen during training, just like in supervised learning. This problem becomes more severe for on-policy RL because the distribution of input data, $(s, c_a)$ used for training $\pi_\theta$ is governed by the actions taken by $\pi_\theta$ itself. This means that the policy can bias its own training data distribution towards a small subset of actions, while ignoring other actions, which could actually be more informative about the actions available at test time. Since there is no prior information on the distribution over action space $\mathcal{A} \subset \mathbb{A}$ at test time, it is assumed to be uniform. Therefore, this discrepancy between training and evaluation due to the non-stationarity of RL training, breaks the identical distribution (in i.i.d.) assumption in statistical learning theory (Bousquet et al., 2003). To address this non-uniformity in training data, the following regularizing techniques are proposed to augment the ERM objective in Eq. 4:

(1) **Maximum entropy regularization**: Maximum entropy objective (Ziebart et al., 2008) augments Eq. 4 with the stochastic policy's entropy $\mathcal{H}[\pi_\theta(a|s)]$ with weight $\beta$, as in Eq. 5. This makes the policy maximize environment reward, under the constraint of taking diverse actions. This helps generalization in two ways: (a) the input data distribution used for training the policy becomes more uniform over action representations, and (b) the policy outputs maximum entropy distributions which make the least assumptions about the possibly unseen set of available actions $\mathcal{A} \subset \mathbb{A}$, and hence by the principle of maximum entropy (Jaynes, 1957; Guiasu & Shenitzer, 1985) overfits the least.

$$\max_\theta \mathbb{E}_{a \sim \pi_\theta(.|s, \mathcal{A}_K)}[R(s, c_a) + \beta \mathcal{H}[\pi_\theta(a|s)]] \tag{5}$$

(2) **Changing action spaces**: The training data distribution can be made more uniform by sampling a set of available actions $\mathcal{A} \subset \mathcal{A}_K$, uniformly in every episode. This blocks certain actions, making the policy select appropriate actions only from the available set $\mathcal{A}$. Hence, the experience collected by the policy is uniformly spread over the known actions $\mathcal{A}_K$, making the training data distribution more identical to the assumed uniform distribution at test time. Eq. 6 shows this training objective:

$$\max_\theta \mathbb{E}_{\mathcal{A} \subset \mathcal{A}_K, a \sim \pi_\theta(.|s, \mathcal{A})}[R(s, c_a) + \beta \mathcal{H}[\pi_\theta(a|s)]] \tag{6}$$

(3) **Clustering similar actions**: The known action space $\mathcal{A}_K$ can contain several groups of similar actions (e.g. various knives for cutting), and a randomly sampled action space $\mathcal{A}$ may contain actions from each group. This can be exploited by the reward-maximizing policy during training to overfit to actions from particular groups, but it will fail to generalize if similar actions are unavailable while testing. To avoid this, we propose to utilize the pre-learned action representations (section 3.2) to partition $\mathcal{A}_K$ into a set of $k$ groups $\mathcal{G}_K = \{g_1 \ldots g_k\}$, where $k$ is a hyperparameter. For every episode during training, an action set $\mathcal{A}_\mathcal{G}$ is built by sampling a subset of groups, $\mathcal{G} \subset \mathcal{G}_K$ and then sampling actions from $\mathcal{G}$ only. Two-step sampling ensures that certain groups of actions are blocked every episode, encouraging the policy to utilize underused action groups as well, making training data more uniform over the action representation space. We use equal-sized variant of $k$-means for clustering. The overall objective is formalized in Eq. 7 below:

$$\max_\theta \mathbb{E}_{a \sim \pi_\theta(.|s, \mathcal{A}_\mathcal{G})}[R(s, c_a) + \beta \mathcal{H}[\pi_\theta(a|s)]], \text{ where } \mathcal{A}_\mathcal{G} \subset \{a|a \in g, g \in \mathcal{G}\} \text{ and } \mathcal{G} \subset \mathcal{G}_K \tag{7}$$

In experiments (Section 5), we perform model selection based on a validation set of actions. We further ablate each regularization techniques and analyze their contribution in different environments.

## 4 ENVIRONMENTS

### 4.1 GRID WORLD

In GRID WORLD environment (Chevalier-Boisvert et al., 2018), an agent navigates a 2D 9x9 maze to reach a goal cell for a sparse reward. A column of lava is randomly placed in every episode, touching which ends the episode. The discrete action space consists of all 5-step macro actions, where each

macro-action is defined by a 5-length sequence of left, right, up or down movement. The entire action space of size $4^5 = 1024$ actions is randomly split into a train and test set of 512 actions. The action datasets are collected on an empty grid where the agent is initialized at random locations. Two kinds of data types are used to represent the state sequence of agent - one-hot vectors and continuous (x,y) grid coordinates.

## 4.2 RECOMMENDER SYSTEM

The RECOMMENDER SYSTEM environment (Rohde et al., 2018) simulates how users may respond to product recommendations. Every episode, the agent must recommend items to a new user with the objective of maximizing the click through rate (CTR) for the recommendations. This simulated environment uses randomly initialized embeddings for recommendations (actions), and we use the same to demonstrate policy generalization to new actions. Action space of size 10,000 is randomly split equally into train and test actions.

## 4.3 CHAIN REACTION TOOL ENVIRONMENT (CREATE)

CREATE is a physics-based puzzle where the goal is to make a specified ball reach a goal position (blue), inspired by the popular video game The Incredible Machine. The agent must place tools in real time to manipulate the path of the ball to reach the goal position. The environment presents a challenging multi-step task, requiring the agent to select the tool to place as well as its position $(x, y)$ on the screen. The agent has access to a subset of diverse tools such as trampolines, see-saws, cannons, funnels, and conveyor belts (Appendix C.2). The position aspect makes this a parameterized action space Hausknecht & Stone (2015) with both discrete and continuous components. Our policy architecture consists of another head to output this continuous vector and it is trained jointly with the discrete action. We solve 3 different CREATE tasks: Push, Navigate and Obstacle. The tools evaluated at test time are completely unseen tool types from those seen during training.

## 4.4 SHAPE STACKING

In SHAPE STACKING the agent must drop blocks on a table to build the highest standing tower. Our objective is different from prior works (Groth et al., 2018; Lerer et al., 2016) in that we maximize the tower height in an RL setting, whereas the prior work predicts the stability of the tower. Similarly to CREATE, the action space in Object Stacking, consists of $(x, y)$ coordinates of where the object should be dropped above the table. This environment is shows our ability to generalize problem solving ability to a new action space in a complex 3D task. The action dataset here are images of the objects from various angles (or viewpoints). In this case the visual appearance of the object is sufficient to infer its functionality.

## 5 EXPERIMENTS

### 5.1 BASELINES & ABLATIONS

**Baselines**: We compare against two policy architectures which can utilize action representations for generalization to unseen action sets. We also compare against a VAE-based non-hierarchical embedding learning method, to learn action representations from unstructured action data (see Fig. 3).

- *Nearest Neighbor*: During training, a policy is learned over all known actions. Given unseen actions, the policy's output is used to select the nearest available action in embedding space.
- *Distance Based*: Based on Dulac-Arnold et al. (2015), a continuous action-space policy outputs in the action embedding space and the closest available action to this output is selected.
- *Non hierarchical VAE*: A shared VAE is trained over the samples across all action datasets. An action's embedding is then computed as the mean over the embeddings of samples in its dataset.

**Ablations**: We individually ablate each of the three proposed regularization metrics in our method.

- *Ours: no entropy*: Trained without entropy regularization by setting entropy coefficient to zero.
- *Ours: no changing*: Trained over the entire set of known actions without any action space sampling.
- *Ours: no clustering*: Training action-space is uniformly sampled (Eq. 6), no $k$-means clustering.

**Alternate embeddings**: We compare how the embedding learning method (HVAE) applies to various forms of unstructured data (Fig. 3). In CREATE, action datasets comprise of state trajectories for

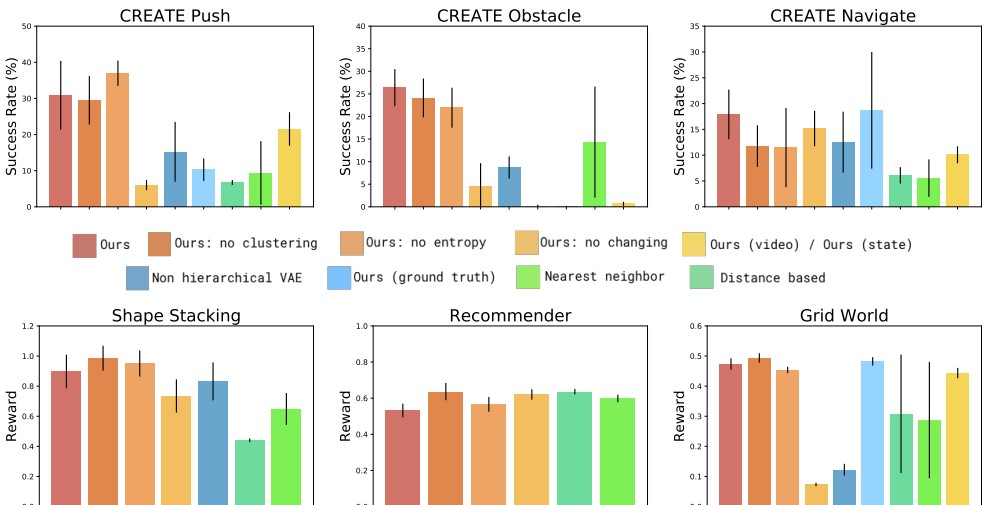

Figure 3: **Quantitative results**: displayed are 3 of the CREATE tasks, the Block Stacking task, the Recommender task and the Grid World task. The performance displayed is measured on generalization to the test set of actions across 3,200 episodes. All results are averaged across 6 seeds. The legend describes ablations of our method (shades of red), embedding baselines (shades of blue), policy architecture baselines (shades of green), and alternate modalities in learning embeddings (yellow).

tool behavior, except *Ours (video)* where video datasets are used instead. In Grid World, action datasets contain trajectories of states in one-hot representation, except in *Ours (state)* where states are real-valued 2D coordinates. *Ours (Ground Truth)* representations are not learned, but instead uses manually engineered representations for the actions. Detailed descriptions are present in Appendix C.

## 5.2 QUANTITATIVE RESULTS

The generalization performance of the policy to unseen actions across all environments and method variations is shown in 3. As seen from the results our method or ablations all of our methods have the strongest ability to generalize to unseen actions across a variety of environments. The difference among our ablations is smaller in simpler environments like Grid World, Recommender systems and Shape Stacking, where the unseen action spaces are very similar to training actions. The effect of clustering-based sampling and entropy regularization can be seen for Obstacle and Navigate environments, which require solving the task with quite different tools at testing. CREATE Push is solvable with a wide variety of tools, and hence the no-entropy policy trains to a higher reward, and is able to generalize as well as many unseen tools can solve the task easily. The performance of our method against its variant with non-hierarchical VAE embeddings shows the importance of hierarchy in latent space to represent actions.

We test the generalizability of our embedder and policy for the task of zero-shot generalization to unseen actions. Specifically, our primary experiments across all four environments, discussed in section 4, train a policy on a fixed set of actions, tune hyperparameters on a separate evaluation set, and then test the ability to generalize to a new set of actions. We further provide qualitative analysis on cases where this generalization succeeds and fails. Finally, we evaluate how our method's generalizability varies with the degree of difference between seen and unseen.

## 5.3 FURTHER ANALYSIS

Qualitative results of the policy test performance are shown in 5. The left and middle column contain success cases. In the left column for CREATE we seen the policy, despite never having used on of the tools before, still be able to solve the task. Likewise, for shape stacking we see the policy able to use novel shapes to build a tall and stable pile to maximize the height. We also show cases of failure

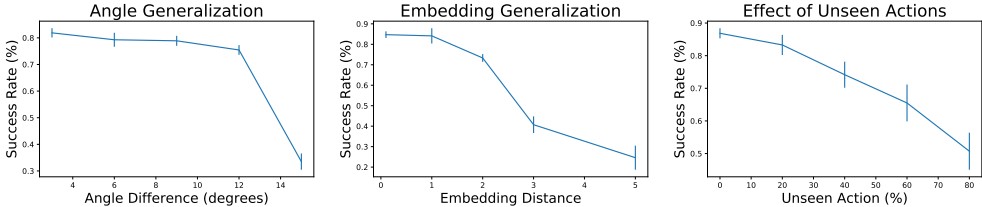

Figure 4: **Varying difficulty of test action space:** (i) Each test action is at least a specific angle apart from all actions seen during training (ii) Each test action is at least a specific distance in embedding space apart from all actions seen during training (iii) Test set contains seen/unseen ratio

to generalize in the right most column. In both cases the policy chooses the right types of actions and barely misses the objective.

We also analyze the conditions needed for generalization to unseen actions. We perform all analyses on CREATE Push task because of the large diversity of tool functionalities. We show generalization across changing physical tool parameters with angle and the embeddings the policy is trained and tested on. Finally, we show the effect of unseen versus seen actions on performance.

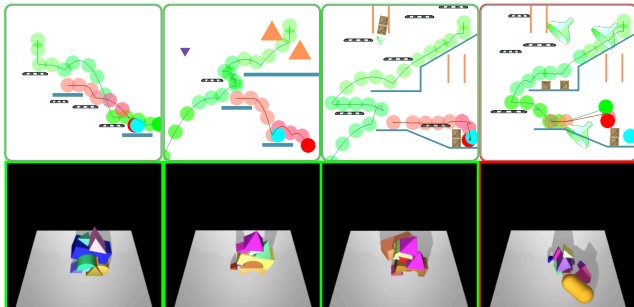

Figure 5: **Qualitative analysis**: shown are two success cases and one failure case for CREATE and Object Stacking. In CREATE the trace of the ball trajectory is outlined. All of the tools or objects in these results the policy is generalizing to select and was not trained over these actions.

## 6 CONCLUSION

Generalization to novel circumstances is an important ability to have, for robust and widely applicable artificial agents. In this paper we propose the problem of generalization of reinforcement learning policies to unseen spaces of actions, with the use of action representations learned in an unsupervised manner. Our two-phase framework demonstrates how representation learning can be combined with the downstream task of reinforcement learning, specifically to represent actions. We demonstrate the efficacy of our methods on four challenging environments, and discuss which variants work when. The key takeaway is that when unseen actions are quite different from known actions, then more regularization helps to train generalizable policies.

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

# A    ALGORITHM

---

**Algorithm 1** Training

---

**Require:** Set of known actions $\mathcal{A}_K = \{a_1, \ldots a_N\}$ , associated action datasets $\{D_1, \ldots D_N\}$, number of clusters $k$ and number of clusters to sample from $S$

   Initialize HVAE model parameters
   **for** epoch = 1, 2, ... **do**
      Sample $a_i \sim \mathcal{A}_K$
      Compute loss in Eq. 1 for $D_i$
      Update HVAE model with gradient of Eq. 1
   **end for**
   Initialize empty set $\mathcal{C}_K = \{\}$
   Infer action representations $c_{a_i} = q_a^{\mu}(c|D_i), \forall a_i \in \mathcal{A}_K$ {$\mu$ denotes mean of distribution}
   Store action representations $\mathcal{C}_K \leftarrow \{c_{a_i} \ldots c_{a_N}\}$
   Initialize policy parameters $\theta$
   Compute $k$-means clustering on $\mathcal{C}_K$: $\mathcal{G}_k \leftarrow \{g_1, \ldots, g_k\}$

   $t \leftarrow 0$
   Receive initial state $s_0$ from ENV and initialize replay buffer
   **while** not done **do**
      **for** step in rollout buffer **do**
         Sample clusters $\mathcal{G} \subset \mathcal{G}_K$
         Sample action set $\mathcal{A}_g$ from clusters $\mathcal{G}$ as in Eq. 7
         Compute $f_\theta(s_t, c_{a_j}), \forall a_j \in \mathcal{A}_g$
         Sample action $a_t \sim \pi_\theta(s)$ using Eq. 2
         $s_{t+1}, r_t \leftarrow \text{ENV}(s_t, a_t)$
         Store experience $(s_t, a_t, s_{t+1}, r_t)$ in replay buffer
         $t \leftarrow t + 1$
      **end for**
      Update $\theta$ based with PPO using rollout buffer
   **end while**

---

---

**Algorithm 2** Testing

---

**Require:** Set of test actions $\mathcal{A}_\mathcal{T} = \{a_{N+1}, \ldots a_{N+M}\}$ and action datasets $\{D_{N+1}, \ldots D_{N+M}\}$

   Infer action representations $c_{a_i} = q_a^{\mu}(c|D_i), \forall a_i \in \mathcal{A}_T$
   **while** not done **do**
      Compute $f_\theta(s_t, c_{a_i}), \forall a_i \in \mathcal{A}_T$
      Sample action $a_t \sim \pi_\theta(s)$ using Eq. 2
      $s_{t+1}, r_t \leftarrow \text{ENV}(s_t, a_t)$
      $t \leftarrow t + 1$
   **end while**

---

HVAE implementation is based on the PyTorch implementation of Neural Stastician (Edwards & Storkey, 2017), and we use RAdam optimizer (Liu et al., 2019). For training our policy, we use PPO (Schulman et al., 2017; Kostrikov, 2018) with Adam optimizer (Kingma & Ba, 2015).

# B ANALYSIS OF TRAINING PROCEDURE AND MODELS

## B.1 VISUALIZATION OF HIERARCHICAL EMBEDDING SPACES

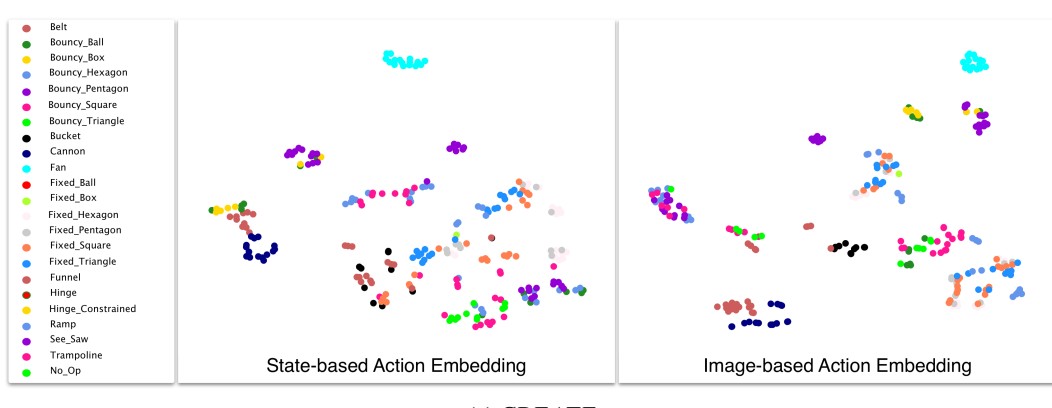

(a) CREATE

Figure 6: T-SNE Visualization of learned embedding space for CREATE environment. Tools in CREATE are labeled by their properties which define them to be floor, trampolines, high-frictional, etc. The action embeddings clearly group similar actions together.

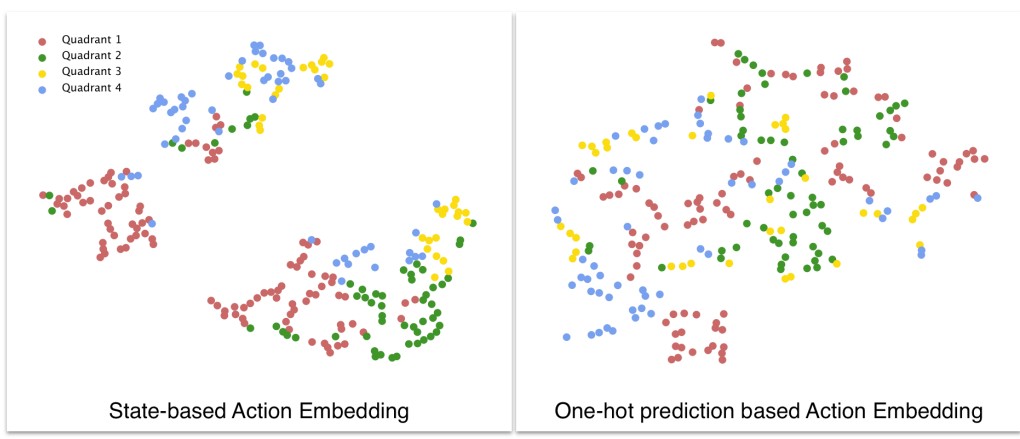

(a) GRID WORLD

Figure 7: T-SNE Visualization of learned embedding space for Grid World environment.

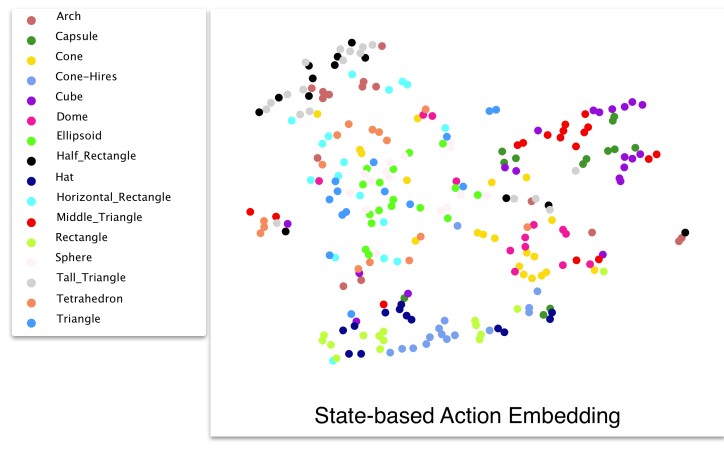

(a) SHAPE STACKING

Figure 8: T-SNE Visualization of learned embedding space for Shape-stacking environment.

## B.2 PERFORMANCE CURVES FOR ABLATIONS

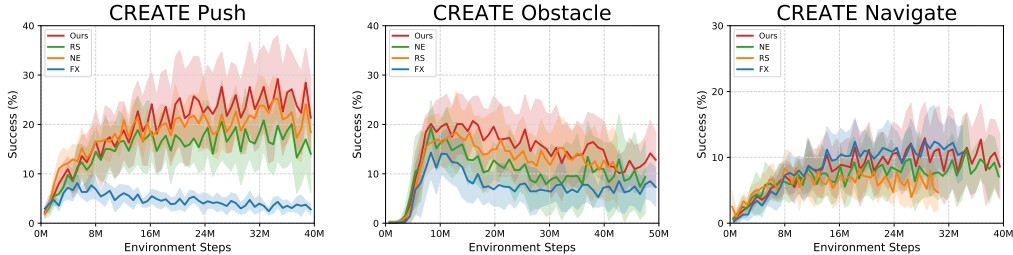

Figure 9: Success rate curves comparing the learning of the primary method *Ours* versus ablations. The success rate shown is being evaluated on the validation action set. Note that performance decreasing on the validation set indicates overfitting to the training set of actions.

## B.3 FINE-TUNING COMPARISON

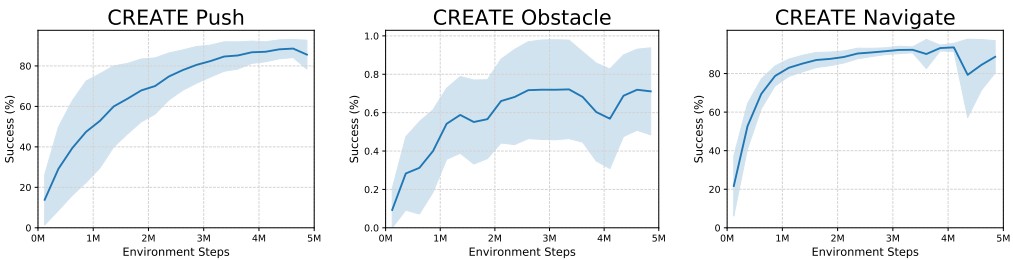

Figure 10: Fine-tuning the policy to adapt to a new action space by re-initializing the final layer. Performance is computed across 5 seeds trained from the same model.

One possible way to adapt a policy to an unseen set of actions is to spend time fine-tuning the learned policy for the new action space. This process keeps the earlier layers of the policy but fine-tunes the actual action selection in the final layer by re-initializing the output layer. This process is able to adapt to an unseen action set of any size. However, this fine-tuning process can be time consuming which is why we try to achieve generalization in this paper. In Figure 10, we show the performance

of such fine tuning on the three CREATE tasks. As seen from the figure, it takes millions of steps to fine-tune, proving prohibitive for adaptive policies. On the other hand our framework generalizes to the new action space without the need for expensive RL retraining.

## C ENVIRONMENT DETAILS

### C.1 GRID WORLD

GRID WORLD environment consists of an agent and a lava wall with an opening as shown in figure 11. The lava wall can be either horizontal or vertical. The agent is spawned in a random position and can move in 4 directions (up, down, left, and right). The objective of the agent is to reach the goal in the bottom-right corner while avoiding lava.

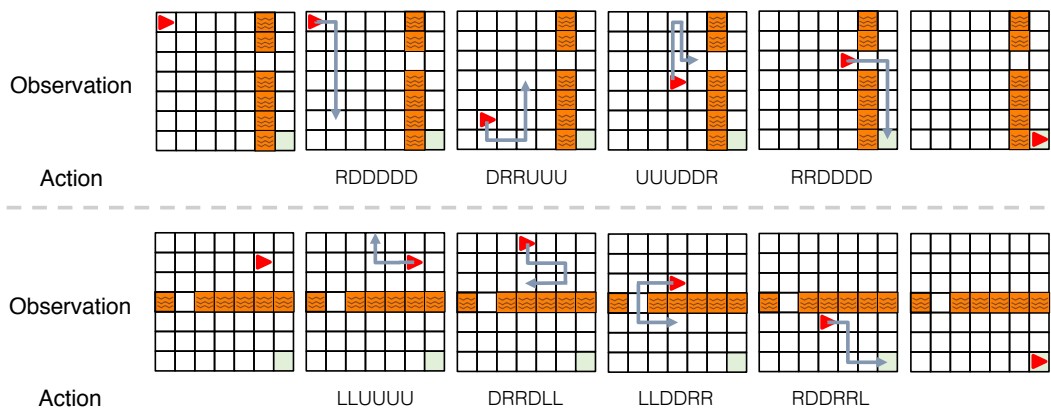

Figure 11: GRID WORLD environment. The agent is the red triangle and the goal is the green cell. The environment contains one row or column lava wall with a single opening. Each action consists of 6 consecutive moves in 4 directions.

**Observations**: The observation space is 9x9x3 where three channels represent (1) object id (agent, wall, goal, lava), (2) color of an object, and (3) zero in each cell. The agent gets a flattened vector of this 9x9x3 matrix.

**Actions**: An action of the agent is 5 consecutive moves in 4 directions. Hence, $4^5 = 1,024$ actions are possible in total. Once the agent selects an action, it executes 5 sequential moves step-by-step. During an action execution, if the agent hits the boundary, it will stay in the current cell. If the agent steps on lava, the game will be terminated. The whole action set is divided into 8:1:1 split of train, validation and test action sets.

**Rewards**: GRID WORLD provides a sparse reward, $1 - 0.9 \times (\text{step/max\_step})$, only when the agent reaches the goal. The reward is discounted based on the number of actions taken to encourage a shorter path to the goal.

**Termination**: Each game is terminated when the agent takes max_steps (64) actions or the agent reaches the goal or the lava wall.

**Action Datasets**: The action datasets are observations of an agent performing actions in a 80x80 grid with no obstacles. The observations constitute the trajectory of states the agent sees during application of a macro-action, starting from randomized initial states. A dataset of 1024 such trajectories is used to represent a single macro-action, and is encoded by the HVAE to an action representation which identifies the underlying action behavior. We consider three kinds of embeddings in our experiments:

- One-hot (default): State is represented by two 80-dimensional one-hot vectors of x and y coordinates of the agent's locations on the 80x80 grid. Autoencoder reconstruction is based on a softmax cross-entropy loss over the states in the trajectory. The learned action embeddings are 16-dimensional.

- State: Two-dimensional continuous vector of x and y coordinates of the agent in the 80x80 grid. Reconstruction loss is computed based on Gaussian log-likelihood over the states in the trajectory. The learned action embeddings are 16-dimensional.

- Ground-truth: These are 5-dimensional embeddings containing the true knowledge of the five moves (up, down, left, right) that constitute a macro-action.

## C.2 CHAIN REACTION TOOL ENVIRONMENT (CREATE)

CHAIN REACTION TOOL ENVIRONMENT (CREATE) is a physics-based puzzle where the objective is to make a target ball reach a goal position by placing variety of tools, inspired by the popular video game "The Incredible Machine". The environment contains two movable objects, a marker ball (green) and a target ball (red). When a game starts, the marker ball is falling off from the top of the screen and an agent requires to place tools to redirect the kinetic energy of the marker ball to the target ball so that the target ball reaches the goal position (blue) as illustrated in Figure C.2.

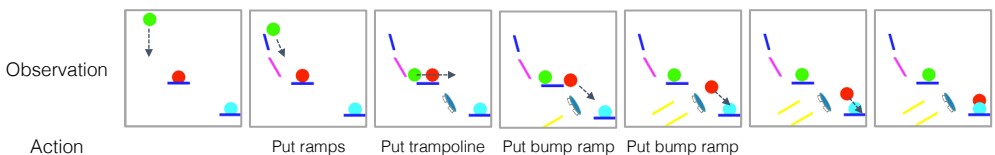

Figure 12: CREATE environment. In CREATE, the green ball is falling into the scene, which must push the red target ball into the blue goal location. The top and bottom rows show actual evaluation results when our model is tested on CREATE UP and CREATE DOWN, respectively.

**Observations**: An observation for each time-step is an 84x84x3 image of a game screen and we use 3 frame stack to provide information about velocity and acceleration of the balls. Initially, the game contains 3 balls in the observation: marker ball (green), target ball (red), and goal ball (blue).

**Actions**: CREATE contains 1,737 tools of type: ramps, trampolines, walls, balls, floor, conveyor belts, funnel, polygons of different shapes, cannons, fans and buckets. We also have an action for *No-Operation*. The whole tool set is divided into train (939), validation (400) and test sets (400). Every time-step, the agent outputs a parameterized action, i.e. a discrete-continuous action which has three values $(tool, x, y)$, where $tool$ specifies which tool to place and $(x, y)$ represents the position of the tool in the screen.

**Rewards**: The agent gets +1 reward when the marker ball hits the target ball, and +10 reward when the target ball passes the goal. In addition to reward for success, the environment provides some intermediate rewards. For every time-step, +0.01 reward is given to encourage the agent to keep balls inside the screen. At the same time, an invalid action is penalized by giving -0.01 reward (e.g., placing a tool outside of the screen or placing a tool on top of other tools).

**Termination**: Each game is terminated when the agent takes 20 actions, or the marker ball goes out of the screen before it hits the target ball, or the target ball goes out of the screen before it passes the goal.

**Action Dataset**: The action dataset is constructed by testing the properties of each tool through scripted interactions with a probe ball. A ball is launched at a given tool from various angles and speeds and then interacts with the tool. The properties of the tool will determine the deflection path of the ball. Testing interaction from various angles and speeds helps to build a better understanding of the tool. The collected datasets consist of 1024 trajectories of length 10 of the ball's interaction with the tool. Each trajectory consists of observations in the form of either environment state (default) or 48x48 gray-scale images (*Ours (image)* in Fig. 3). We consider three kinds of embeddings in our experiments:

- State trajectory (default): Each environment state in a trajectory is represented by 2D coordinates and 2D velocity of the probe ball, concatenated with 2D tool's location. Autoencoder reconstruction is based on Gaussian log-likelihood over the states in the trajectory. The learned action embeddings are 128-dimensional.

- Video: Each state in a trajectory is represented by 48x48 gray-scale image, which makes the trajectory a video of probe ball's interaction with the tool. Autoencoder reconstruction is based on Gaussian log-likelihood over the images in the input video. The learned action embeddings are 128-dimensional.

- Ground-truth: These are 32-dimensional embeddings containing the true knowledge of the tool in the form of a one-hot encoding of the tool type along with its properties such as angle, length, elasticity, etc.

### C.3    SHAPE STACKING

SHAPE STACKING is a mujoco simulation environment (Todorov et al., 2012) where the agent is given a set of objects of different shapes of varying sizes, for instance, cubes, rectangles, spheres, round cylinders, archs, etc. The objective is to stack a tower as high as possible by choose the appropriate objects given in a particular episode.

**Observations**: An observation for each time-step is an 84x84x3 image of the shapes laying on the table.

**Actions**: In total there are 900 distinct shapes in the environment. 675 are used for learning the policy. The remaining 225 are used for evaluating performance on unseen actions. The same types of polygons do not appear in both train and test. Like CREATE the action space includes making a discrete selection over the shape to drop and the $x, y$ coordinates to drop the shape above the scene.

**Rewards**: The agent receives a reward for the positive difference in height of the tower. A penalty of $-0.25$ is given for every repeated shape.

**Termination**: The game is terminated either after the tower of shapes exceeds 3 in height or after 10 shape placements.

**Action Dataset**: In Shape Stacking the functionality of each action is characterized by the physical appearance of the shape. For this reason, the action dataset consists of images of the shape from various angles and heights (viewpoints). 1,024 images of resolution 84x84 constitute the dataset to represent a single shape.

