# OpenReview forum: "Generalizing Reinforcement Learning to Unseen Actions"
_ICLR.cc/2020/Conference — Reject_

### Official Review · AnonReviewer2 · 2019-10-23
**Official Blind Review #2**

**Rating:** 6

**Review:**

This paper addresses the very interesting problem of generalising to new actions after only training on a subset of all possible actions. Here, the task falls into different contexts, which are inferred from an associated dataset (like pixel data). Having identified the context, it is used in the policy which therefore has knowledge of which actions are available to it.

In general, this paper is written very well, with some minor suggested tweaks in the following paragraphs. The key strategies used here (HVAE to identify contexts, ERM as the objective, entropy regularisation, etc) all make sense, and are shown to work well in the experiments carried out.

While the experiments are sufficiently varied, it worries me that only 3 or 2 seeds were used. In some cases, such as NN and VAE in the CREATE experiments show large variances in performance. Perfects a few more seeds would have been nice to see. This is the key reason why I chose a 'Weak Accept' instead of an 'Accept'.

Some of the results (the latent spaces) shown in the appendix are very interesting too, particularly since they show how similar actions spaces cluster together in most cases.

Minor issues:

1) In Figure 3, I am not clear about what 'im' and 'gt' settings are.
2) In Figure 3, it would have been nice to have consistent colors for the different settings.
3) It would have been nice to see the pseudocode of the algorithm used.


**Experience Assessment:**

I have read many papers in this area.

**Review Assessment: Checking Correctness Of Derivations And Theory:**

N/A

**Review Assessment: Checking Correctness Of Experiments:**

I assessed the sensibility of the experiments.

**Review Assessment: Thoroughness In Paper Reading:**

I read the paper at least twice and used my best judgement in assessing the paper.

---

> ### Author Response · Authors · 2019-11-15
> **Response to Reviewer #2**
>
> We thank the reviewer for their valuable feedback and time. We have updated the style of experiments section to address all the suggested improvements:
>
> 1. Additional seeds in results.
> We have updated all the experimental results displayed in Figure 3, to be evaluated over 6 seeds. In Figure 9 (Appendix B.2) we have also added success rate curves, showing variation across different seeds as training progresses, for our method and ablations in the CREATE environment.
>
> 2. Definition of "im" and "gt" settings.
> In response to the reviewer’s feedback, we have renamed the "im" setting to "Ours (video)" and "gt" setting to "Ours (ground truth)" for clarity. We have also added a detailed explanation for these settings as well as other baselines and ablations in the revised Section 5.1 "Baselines and Ablations". Description of these settings are:
> (a) "Ours (video)": This setting is for the CREATE environment, where the action dataset used for learning embeddings is composed of videos (sequence of image-based environment states). By default, the CREATE results on "Ours" are based on environment state trajectories.
> (b) "Ours (ground truth)": shows the performance of our method with manually engineered action embeddings for CREATE and Grid world environments.
> We have added comprehensive details for various alternate embeddings that were tested, under the "Action Dataset" subsections in Appendix C for each environment.
>
> 3. Recolored Figure 3.
> We thank the reviewer for the helpful comments on presentation. We have updated the colors in Figure 3 to be consistent across environments, and also clearly delineate the color codes for our method, ablations (shades of red), embedding-method baselines (shades of blue), policy architecture baselines (shades of green), and alternate kinds of learned embeddings (yellow).
>
> 4. Algorithm pseudocode.
> We have added pseudocode of our training and testing algorithm to appendix section A.

---

### Official Review · AnonReviewer3 · 2019-10-23
**Official Blind Review #3**

**Rating:** 6

**Review:**

This paper studies the problem of generalization of reinforcement learning policies to unseen spaces of actions. To be specific, the proposed model first extracts actions’ representations from datasets of unstructured information like images and videos, and then the model trains a RL policy to optimize the objectives based on the learned action representations. Experiments demonstrate the effectiveness of the proposed model against state-of-the-art baselines in four challenging environments. This paper could be improved in the following aspects:
1.	The novelty of the proposed model is somewhat incremental, which combines some existing methods, especially the unsupervised learning for action representation part that just combines methods such as VAE, temporal skip connections…
2.	Some components of the proposed methods are ad hoc, and are not explained why using this design, such as why Bi-LSTM for encoder and why LSTM for decoder.
3.	More definitions about the model should be offered, such as “y^∗ is some optimal action distribution”, how to get the optimal action distribution?
4.	Some datasets are not sufficient enough for sake of statistical sufficiency, such as recommendation data with only 1000 action space.
5.	The contributions of action regularizations are not validated on experiment section.


**Experience Assessment:**

I have published in this field for several years.

**Review Assessment: Checking Correctness Of Derivations And Theory:**

I carefully checked the derivations and theory.

**Review Assessment: Checking Correctness Of Experiments:**

I carefully checked the experiments.

**Review Assessment: Thoroughness In Paper Reading:**

I read the paper thoroughly.

---

> ### Author Response · Authors · 2019-11-15
> **Response to Reviewer #3 (Part 2)**
>
> 2. Reasoning for proposed method design choices.
> We have incorporated the reviewer’s helpful comments by adding more references and explanations for various design choices in the method section:
>
> - Trajectory autoencoders:
> In Section 3.2, we follow prior work on trajectory autoencoders [5, 6] for the choice of using Bi-directional LSTM encoder [7] and LSTM decoder. Our work extends hierarchical VAE to trajectory settings, when the action datasets are composed of state trajectories or videos. We have added these references in the paper.
>
> - Using distribution mean as representation:
> In section 3.2, we have added justification for using the encoder’s output mean as an action representation by referencing prior work in representation learning like [8, 9]. We further note that the encoder’s output distribution (mean and standard deviation) can also be used as a representation, as done in [10].
>
> - Design choices for enabling generalization:
> We have revised section 3.4 to explain the components of regularization in detail, and added suitable references to justify design choices. Specifically, we add details for connection with statistical learning theory, discuss the need for regularization metrics, revise justification and add references for the principle of maximum entropy, and methodically develop each regularization metric.
>
>
> 3. More definitions about model.
> We have added detailed explanations, definitions and references in our revision to Section 3.4 on enabling generalization in RL. We clearly redefine each term used in the equations, such as optimal action distribution y^* for a given state, loss function L measuring the optimality of a policy. The proposed regularization metrics are defined in mathematical terms, and overall training objective is added to Equation 7. We also added Algorithm pseudocode in Appendix A for summarizing the method (suggested by reviewer 2).
>
>
> 4. Statistical sufficiency in action datasets.
> We have increased the number of actions in the recommender environment to 10,000, re-ran all experiments for this environment, and updated the results in Figure 3. For reference, sizes of action space in other environments: Grid World has 1024 macro-actions, CREATE has 1,739 tools and Shape-Stacking has 900 shapes. Furthermore, we increased the number of seeds for all experiments to be 6 for more statistically significant results (suggested by reviewer 2).
>
>
> 5. Validating action regularization contributions.
> Each of the proposed regularization metric’s contribution can be seen in Figure 3 (shades of red are these ablations). In summary, having entropy-regularization and changing-action-space contribute most to the performance while action-clustering can boost performance on challenging environments such as CREATE Navigate and Obstacle.
>
> We thank the reviewer for pointing out the need for further clarity in the action regularization contributions. While the original submission contains ablation studies for each regularization metric, notated with "NE", "FX" and "RS", we have renamed these ablations for better clarity:
> "Ours: no entropy": without maximum entropy regularization (previously NE).
> "Ours: no changing": without changing action spaces regularization (previously FX)
> "Ours: no clustering": without clustering similar actions (previously RS).
> We have also revised section 5.1 "Baselines and Ablations" and Figure 3 to clearly define each compared method. For further analysis of regularization metrics, we have added Figure 9 in Appendix section B.2 which compares success rate curves for ablations.
>
>
> References
> [5] Wang, Ziyu, et al. "Robust imitation of diverse behaviors." Advances in Neural Information Processing Systems. 2017.
> [6] Co-Reyes, John, et al. "Self-Consistent Trajectory Autoencoder: Hierarchical Reinforcement Learning with Trajectory Embeddings." International Conference on Machine Learning. 2018.
> [7] Schuster, Mike, and Kuldip K. Paliwal. "Bidirectional recurrent neural networks." IEEE Transactions on Signal Processing 45.11 (1997): 2673-2681
> [8] Higgins, Irina, et al. "beta-VAE: Learning Basic Visual Concepts with a Constrained Variational Framework." ICLR 2.5 (2017): 6.
> [9] Steenbrugge, Xander, et al. "Improving generalization for abstract reasoning tasks using disentangled feature representations." arXiv preprint arXiv:1811.04784 (2018).
> [10] Locatello, Francesco, et al. "Challenging Common Assumptions in the Unsupervised Learning of Disentangled Representations." International Conference on Machine Learning. 2019.

---

> ### Author Response · Authors · 2019-11-15
> **Response to Reviewer #3 (Part 1)**
>
> We thank the reviewer for their valuable feedback and time. We have revised the presentation and explanations in the method and experiments section to address all the suggested improvements. We address each concern in detail below:
>
>
> 1. Explanation of model novelty.
> The novelty of our proposed method includes (a) learning representations of discrete actions from task-agnostic behavioral datasets such as trajectories and videos, (b) training stochastic policies which can extract task-specific information from learned action representations, (c) formalizing generalization over actions in RL and developing regularization techniques to avoid overfitting. We describe these contributions in detail below:
>
> (a) Learning action representations from behavioral datasets
> Since actions can have diverse behaviors which cannot be explained by a single datapoint, we propose to represent actions with datasets of behaviors. We propose using a Hierarchical VAE, and demonstrate how hierarchy in the VAE leads to better representations of action datasets for downstream tasks in RL (Figure 3: comparison with non-hierarchical VAE).
> Also, the combination of trajectory autoencoders with Hierarchical VAE (HVAE) is novel for representation learning, to the best of our knowledge. We demonstrate representation learning on high dimensional trajectory datasets like videos (Figure 3: Ours (video) experiments).
>
> (b) Training stochastic policies over action representations
> As mentioned in related work (Section 2), some prior works have utilized action representations in Q-learning form to learn deterministic policies [1, 2] and some train actor-critic continuous policies to output in the action embedding space and select nearest neighbor action [3, 4]. In contrast, we propose to do this by defining a utility function over each available action, and then train a stochastic policy with policy gradients through a softmax over this utility function (Equation 2 and Section 3.3). In Figure 3, our proposed policy architecture is shown to outperform "Distance based" method (analogous to [3,4]), using the same learned representations, whereas Q-learning methods like [1, 2] do not learn stochastic policies.
>
> (c) Formalizing action space generalization in RL
> Thanks to the reviewer’s comments, we have revised section 3.4 to justify and exposit the novelty of our framework for generalizing over action space. Our core contribution is proposing regularization techniques to enable generalization to unseen actions in RL, by drawing a connection from statistical learning theory. We discuss how iid assumptions necessary for generalization are violated in on-policy RL, and propose 3 ways to make the training data distribution to be more uniform over known actions. Figure 3 validates the experimental contributions of each of these proposed metrics in various environments.
>
> We would like to emphasize that this paper proposes a novel problem of generalization over action space in RL, demonstrates a combination of unsupervised representation learning with reinforcement learning as a downstream task, and provides two new environments, CREATE and Shape-Stacking, for benchmarking performance on unseen action spaces.
>
> References
> [1] He, Ji, et al. "Deep reinforcement learning with a natural language action space." arXiv preprint arXiv:1511.04636 (2015).
> [2] Tennenholtz, Guy, and Shie Mannor. "The Natural Language of Actions." International Conference on Machine Learning. 2019.
> [3] Van Hasselt, Hado, and Marco A. Wiering. "Using continuous action spaces to solve discrete problems." 2009 International Joint Conference on Neural Networks. IEEE, 2009.
> [4] Dulac-Arnold, Gabriel, et al. "Deep reinforcement learning in large discrete action spaces." arXiv preprint arXiv:1512.07679 (2015).

---

### Official Review · AnonReviewer1 · 2019-10-25
**Official Blind Review #1**

**Rating:** 3

**Review:**

This paper deals with the problem of how to enable the generalization of discrete action policies to solve the task using unseen sets of actions. The authors develop a general understanding of unseen actions from their characteristic information and train a policy to solve the tasks using the general understanding. The challenge is to extract the action's characteristics from a dataset. This paper presents the HVAE to extract these characteristics and formulates the generalization for policy as the risk minimization.

Strengths:
1. This paper shows us how to represent the characteristics of the action using the a hierarchical VAE.
2. From the provided videos, we can directly observe the results of this model applied to different tasks.

Weaknesses:
1. In the paper, the authors mentioned that they proposed the regularization metrics. However, they didn't describe them in details. It is important to develop the proposed method in theoretical style.
2. Analyzing the regularization metrics should be careful in the experiments.
3. Since there are many previous works related to action representation, the experiments should contain the comparison with the other method to see how much improvement was obtained.

**Experience Assessment:**

I have published one or two papers in this area.

**Review Assessment: Checking Correctness Of Derivations And Theory:**

I assessed the sensibility of the derivations and theory.

**Review Assessment: Checking Correctness Of Experiments:**

I did not assess the experiments.

**Review Assessment: Thoroughness In Paper Reading:**

I made a quick assessment of this paper.

---

> ### Author Response · Authors · 2019-11-15
> **Response to Reviewer #1**
>
> We thank the reviewer for their valuable feedback and time. We have made several changes to the presentation of the method and experiment section to address the reviewer’s concerns. We respond to each concern below:
>
> 1.  Details of regularization metrics
>
> To improve the presentation of regularization metrics, we have revised the method section in the paper. Section 3.4 now provides detailed explanations, references for each design choice and methodically develops the regularization metrics.
>
> Summary of Section 3.4 revision: Equation 3 draws the connection with statistical learning theory and applies it to Reinforcement Learning (RL). We discuss how the iid assumptions are violated in on-policy RL which can hurt generalization. Equation 4 describes how usual reward maximization objective is analogous to Empirical Risk Minimization. We revise details of why regularization is needed and develop the proposed metrics with clear mathematical definitions and detailed explanations. Equations 5, 6, and 7 walk through how each of the proposed metrics modify the training objective to enable better generalization at test time.
>
>
> 2. Analyzing regularization metrics in experiments.
>
> The contribution of each proposed regularization metric can be seen in Figure 3 (shades of red are these ablations). In summary, having entropy-regularization and changing-action-space contribute most to the performance while action-clustering can boost performance on challenging environments such as CREATE Navigate and Obstacle (detailed discussion in Section 5.2)
>
> We thank the reviewer for pointing out the need for further clarity in the experiment section. We note that the original submission contains these ablation studies for each regularization metric, notated with "NE", "FX" and "RS". We have defined these ablations clearly (Section 5.1) and renamed them for better clarity:
> "Ours: no entropy": without maximum entropy regularization (previously NE).
> "Ours: no changing": without changing action spaces (previously FX)
> "Ours: no clustering": without clustering similar actions (previously RS).
> For further analysis of regularization metrics, we have added Figure 9 (Appendix B.2) which compares success rate curves for ablations.
>
>
> 3. Comparisons against prior work on action representations.
> To the best of our knowledge, it is hard to directly apply other prior related works on action representation to the proposed problem of this paper (i.e. generalization to unseen actions).
>
> The followings describe why prior work on action representations is not well-suited (also described in Section 3):
> - [1] assumes access to action embeddings and proposes training continuous policies whose output vector is used to select the closest action embedding. The "Distance-based" method (Section 5.1 and Figure 3) is analogous to [1], and we tailor it for the problem of generalization by comparing nearest action embeddings only from the set of available actions. Our method far outperforms this baseline by extracting task-specific information from the action embeddings as input.
> - While [2] deals with the problem of learning action representations, it is not suitable for generalization to unseen actions. This is because the method requires a fixed discrete action space to learn action representations implicitly, and hence cannot accommodate any new actions without a retraining period.
> - While [3] proposes to pre-learn action representations, the method requires task-specific demonstrations, which reflect the co-occurrence of actions while solving certain tasks. This cannot be extended to unseen actions, as it is not reasonable to assume task-specific demonstrations for them.
>
> Other baseline methods, not directly associated with any prior work, are described in Section 5.1. Among action representation baselines, we compare against a non-hierarchical VAE model - to assess the importance of hierarchy in learning action representations from datasets. We also compare performance against manually engineered ground truth action representations.
>
> Additionally, following the insights of the reviewer to further validate the improvements due to our method, we performed an additional experiment on fine-tuning a well-trained discrete action policy. We show how long it takes to retrain the policy when the final layer is re-initialized for the unseen action set. The results in Figure 10 (Appendix B.3), show that it can take hundreds of thousands of environment steps to achieve performance similar to our method which directly generalizes zero-shot to any new action set.
>
> References
> [1] Dulac-Arnold, Gabriel, et al. "Deep reinforcement learning in large discrete action spaces." arXiv preprint arXiv:1512.07679(2015).
> [2] Chandak, Yash, et al. "Learning action representations for reinforcement learning." arXiv preprint arXiv:1902.00183(2019).
> [3] Tennenholtz, Guy, and Shie Mannor. "The natural language of actions." arXiv preprint arXiv:1902.01119 (2019).

---

### Author Response · Authors · 2019-11-15
**Paper Revision Summary**

We would like to sincerely thank all the reviewers for their constructive comments. We have revised our paper to incorporate them. The updates are summarized as follows:

1. [Approach Section] Revised details and definitions on regularization metrics
The revision to method section (Section 3.4) mathematically draws a connection between statistical learning theory and our problem of generalization in RL. For each proposed regularization metric, we have added elaborate explanations, references for design choices, and training objectives. (Reviewer #1, Reviewer #3)

2. [Experiments Section] Revised presentation of ablation studies
- For easier understanding of comparison with regularization metrics, we have renamed all the compared methods to have complete names with descriptions (Section 5.1). (Reviewer #1, #3)
- Figure 3 is revised to have consistent colors for ablations (red shades), embedding-related baselines (blue shades) and policy-related baselines (green shades) - across all environments. (Reviewer #2)

3. [Experiments] Evaluation on more seeds, more actions for Recommender environment
- All experiments have been evaluated to have a total of 6 seeds and updated in Figure 3. (Reviewer #2)
- We increased the recommender system action space size from 1,000 to 10,000 for statistical sufficiency. (Reviewer #3)

4. [Appendix] Additional analysis: How slow is fine-tuning compared to zero-shot generalization?
Our proposed method generalizes zero-shot to unseen actions by utilizing action datasets, and prevents expensive RL retraining. In contrast, it takes hundreds of thousands of environment steps to fine-tune a standard RL policy to achieve similar performance (Figure 10, Appendix B.3). This further validates improvements due to our method. (Reviewer #1)

5. [Appendix] Additional details on: Pseudocode, Performance curves, Embedding modalities
- Added pseudocode (Appendix A) summarizing training and testing procedure. (Reviewer #2)
- Added performance curves for better analysis of the regularization metrics. (Reviewer #1, Reviewer #3)
- Added details on various data modalities (Appendix C) used for embeddings: trajectories of states, images, videos, ground-truth. (Reviewer #2)

---

### Decision · Program_Chairs · 2019-12-19

**Decision:**

Reject

**Comment:**

This paper proposes a method for reinforcement learning with unseen actions.  More precisely, the problem setting considers a partitioned action space.  The actions available during training (known actions) are a subset of all the actions available during evaluation (known and unknown actions).  The method can choose unknown actions during evaluation through an embedding space over the actions, which defines a distance between actions. The action embedding is trained by a hierarchical variational autoencoder. The proposed method and algorithmic variants are applied to several domains in the experiments section.

The reviewers discussed both strengths and weaknesses of the paper.  The strengths described by the reviewers include the use of the hierarchical VAE and the explanatory videos.  The primary weakness is the absence of sufficient detail when describing the solution.  The solution description is not sufficiently clear to understand the details of the regularization metrics.  The details of regularization are essential when some actions are never seen in training.  The reviewers also mentioned that the experiment analysis would benefit from more care.

This paper is not ready for publication, as the solution methods and experiments are not presented with sufficient detail.